# GnRH Analogues as a Co-Treatment to Therapy in Women of Reproductive Age with Cancer and Fertility Preservation

**DOI:** 10.3390/ijms23042287

**Published:** 2022-02-18

**Authors:** Georgios Valsamakis, Konstantinos Valtetsiotis, Evangelia Charmandari, Irene Lambrinoudaki, Nikolaos F. Vlahos

**Affiliations:** 1Second University Department of Obstetrics and Gynecology, Aretaieion University Hospital, Athens Medical School, Ethnikon and Kapodistriakon University of Athens, 115 28 Athens, Greece; k.valtetsiotis@gmail.com (K.V.); ilambrinoudaki@hotmail.com (I.L.); nfvlahos@gmail.com (N.F.V.); 2First University Department of Paediatrics, Aghia Sophia Childrens Hospital, Athens Medical School, Ethnikon and Kapodistriakon University of Athens, 152 33 Athens, Greece; evangelia.charmandari@googlemail.com

**Keywords:** GnRH analogues, fertility preservation, cancer

## Abstract

In this review, we analyzed existing literature regarding the use of Gonadotropin-releasing Hormone (GnRH) analogues (agonists, antagonists) as a co-treatment to chemotherapy and radiotherapy. There is a growing interest in their application as a prophylaxis to gonadotoxicity caused by chemotherapy and/or radiotherapy due to their ovarian suppressive effects, making them a potential option to treat infertility caused by such chemotherapy and/or radiotherapy. They could be used in conjunction with other fertility preservation options to synergistically maximize their effects. GnRH analogues may be a valuable prophylactic agent against chemotherapeutic infertility by inhibiting rapid cellular turnover on growing follicles that contain types of cells unintentionally targeted during anti-cancer treatments. These could create a prepubertal-like effect in adult women, limiting the gonadotoxicity to the lower levels that young girls have. The use of GnRH agonists was found to be effective in hematological and breast cancer treatment whereas for ovarian endometrial and cervical cancers the evidence is still limited. Studies on GnRH antagonists, as well as the combination of both agonists and antagonists, were limited. GnRH antagonists have a similar protective effect to that of agonists as they preserve or at least alleviate the follicle degradation during chemo-radiation treatment. Their use may be preferred in cases where treatment is imminent (as their effects are almost immediate) and whenever the GnRH agonist-induced flare-up effect may be contra-indicated. The combination treatment of agonists and antagonists has primarily been studied in animal models so far, especially rats. Factors that may play a role in determining their efficacy as a chemoprotective agent that limits gonadal damage, include the type and stage of cancer, the use of alkylating agents, age of patient and prior ovarian reserve. The data for the use of GnRH antagonist alone or in combination with GnRH agonist is still very limited. Moreover, studies evaluating the impact of this treatment on the ovarian reserve as measured by Anti-Müllerian Hormone (AMH) levels are still sparse. Further studies with strict criteria regarding ovarian reserve and fertility outcomes are needed to confirm or reject their role as a gonadal protecting agent during chemo-radiation treatments.

## 1. Introduction

The incidence of cancer in women of a reproductive age remains high. In 2020, breast cancer became the leading type of cancer worldwide, with 2.3 million new cases and 685,000 deaths [1]. In Australia, the cancer incidence rate for women under 40 was 64.7 per 100,000 in 2017 [2]. In the UK, the mean cancer incidence for women under 40 was 56.9 per 100,000 [3]. The International Agency for the Research of Cancer estimates that globally in 2020 there were about 1,380,000 new cancer cases in women under 45 with a 52.2 per 100,000 cancer incidence rate [4].

Depending on age and treatment choices, 15–50% of pre-menopausal women may be expected to develop premature ovarian failure (POF) [5]. This is true especially when chemotherapy is administered for breast cancer and Hodgkin’s Lymphoma (HL) [6,7], though it is also encountered in the treatment of other malignancies such as ovarian and endometrial cancer [8]. Presently, women hope for fertility later in life, and a significant proportion of them have not completed their family at the time of diagnosis. Infertility is an important long-term effect of cancer treatment, especially given the fact that surviving cancer does not seem to drop the desire for childbearing and may increase value placed on familial bonds, though anxieties about potential infertility remain [9,10].

Treatment protocols for cancer patients often include chemotherapy and radiation therapy, both of which are associated with gonadotoxicity, which may result in POF or infertility. POF is caused by apoptosis of primordial follicles and a subsequent loss of ovarian reserve [11]. Alkylating agents are the most toxic, though treatment duration and cumulative dose also plays an important role [7]. Radiotherapy, when targeted to the pelvis, abdomen, or head (by adversely affecting the hypothalamic-pituitary-adrenal axis [12]) can also be gonadotoxic [13]. Past studies showed that ovarian function was preserved in over 90% of long-term female survivors who were treated for lymphoma before puberty, but only in a minority of similarly treated adult patients [14]. The mechanisms behind the toxicity are multiple, such as direct ovarian toxicity through apoptosis of the oocytes, as well as oxidative stress and decreased ovarian blood flow [11].

Due to the treatment’s gonadotoxicity, premenopausal patients are advised to seek fertility preservation, as is the official recommendation of all the cancer such as ASCO [15] and NCCN [16]. Patients would have a range of choices when it comes to fertility options once cancer treatment is imminent. Depending on age, treatment choice, and type of cancer, the patient should be informed of their options by a fertility specialist. They may elect to cryopreserve oocytes, embryos, cryopreserve ovarian tissue itself, transpose the ovaries, and use GnRH analogues (agonists and antagonists) [17,18]. These treatments may be used in combination. This applies especially to the use of GnRH analogues, which may be used either as part the ovarian stimulation protocols or as a chemoprotective agent for ovarian function preservation. They could be used alongside other, non-pharmaceutical, fertility preservation procedures.

The primary issue with most fertility treatments is, however, that they require several days to be completed. Cryopreservation of oocytes can be used as fertility preservation method for women after menarche without a partner [15], with embryo cryopreservation also being a choice for those partnered, or for those wishing to use a sperm bank, and where legally allowed. For oocyte collection, patients may seek in vitro maturation, an experimental procedure [19]. It allows for the immediate collection of immature oocytes, valuable to cancer patients that cannot undergo hormone treatment or delay chemotherapy.

Ovarian tissue cryopreservation after removal by laparoscopic surgery is the only option for young prepubertal females and patients who cannot undergo ovarian stimulation [20]. An experimental surgical method for fertility preservation is transposition of the ovaries outside the radiation field. According to reports ovarian function is preserved in 20% to 100% of patients [21], though we still do not have definitive clinical trials on the efficacy and safety of the procedure.

Presently, GnRH analogues, consisting of agonists and antagonists, are used for fertility preservation. ESHRE recommends ovarian stimulation in women seeking fertility preservation for medical reasons the usage of GnRH antagonist protocol and they further add that there is moderate quality evidence of the necessity of considering a specific GnRH analogue protocol. They state that GnRH antagonist protocols are preferred, since they shorten the duration of ovarian stimulation, offer the possibility of triggering final oocyte maturation with GnRH agonist in the case of high ovarian response, and reduce the risk of ovarian hyperstimulation syndrome. Data on live births are extremely scarce, in particular in cancer patients with vitrified oocytes [22]. ASRM recommends that “GnRH analogues may be used “off label for fertility preservation” [23]. They also state that GnRH agonists may be offered to breast cancer patients to reduce the risk of premature ovarian insufficiency [24] but should not be used in place of other fertility preservation alternatives [15] and that more studies are required to establish the efficacy of this treatment and to determine which patients are the best candidates for its use. According to the National Comprehensive Cancer Network (Guidelines Version 2.2022) GnRH agonists are not considered a form of fertility preservation [25] (Table 1).

As chemotherapy mostly affects tissues with rapid cellular turnover, such as the growing follicles [26], it is hypothesized that gonadotoxicity is lower in prepubertal girls than adult women [27]. Recently, evidence shows that GnRH analogues, by inhibiting the stimulation of gonadotrophins and thus ovarian cellular turnover, could decrease the chance of cellular destruction during gonadotoxic cancer treatments [28], although other mechanisms are also at play. Indeed, GnRH analogues have displayed a decrease in the incidence of POF compared to control and despite a growing interest in them, their long-term effects remain understudied [29]. The aim of this narrative review is to summarize and critically appraise the available data on the potential gonadotoxicity-reducing role of the use of GnRH agonists and antagonists during chemo-radiation therapy for women of reproductive age.

## 2. Methodology

The literature search was performed using the databases of Medline and Scopus. We searched for the phrase “fertility preservation” in combination (using AND as a conjunction) with: “woman reproductive age” (288 combined results), “Hodgkin’s lymphoma” (52), “gynecological cancer” (31), “breast cancer” (593), “AMH” (150), “GnRH agonists” (53) and “GnRH antagonists” (5). We searched for animal and human studies, up to those published by October 2021. From the numerous studies found we kept the meta-analyses, RCTs, prospective studies, retrospective studies, and cohort studies, for an analysis of 37 articles.

## 3. GnRH Analogues; Agonists and Antagonists

### Mechanisms of Action—Physiology

The two types of analogues act through different pathways to produce a similar decrease in GnRH secretion. Agonists, such as Buserelin and Triptorelin [30], take advantage of Gonadotropin-releasing hormone receptor (GnRHR) down-regulation that occurs in chronic GnRH surges, by increasing GnRH secretion. They exert their effect by competitively binding to GnRHR while having a higher affinity and lower enzymatic degradation than GnRH. The GnRHR are desensitized to both the exogenous (analogue) and endogenous GnRH, as the receptor is internalized through receptor-mediated endocytosis [30]. This process is known as homologous desensitization, meaning the attenuation is caused by the agonists on their target receptors. Initially, this creates a flare-up of gonadotrophin production until the receptors down-regulate, which in the long-term inhibits gonadotrophin secretion. GnRH agonistic analogues have two distinct differences from GnRH. In the GnRH agonistic decapeptides, the glycine in position 6 is substituted for hydrophobic groups, as this is the primary site of degradation. Many of them also have a deletion of the glycine in position 10, with an ethyl-amide group substituting the C-terminal [30,31], making them nonapeptides. This increases their affinity to GnRHR. The combined effects of a higher affinity and lower degradation make them two hundred times more potent than endogenous GnRHR [31]. They have a couple of disadvantages; they have a flare-up effect, are contraindicated in estrogen receptor positive breast cancer, reduce bone mass in >6-month treatments, and require an administration of minimum one week pre-chemotherapy [17]. GnRHas are administered every four weeks starting 1 to 2 weeks before the initial chemotherapy dose and are usually continued until the end of the chemotherapy regimen. Some protocols, in order to prevent a flare up produced by GnRHa, add an GnRH antagonist at the initial phase followed by agonist protocol treatment, especially if an early start of chemotherapy is needed [32].

Antagonists, such as Ganirelix and Cetrorelix [30], bind competitively to GnRHR preventing pituitary stimulation and the release of gonadotrophins [33]. GnRH antagonists have a higher number of substitutions than the two found in agonists. They exhibit substitutions in positions 1–3, 6, 8 and 10 [30], remaining decapeptides. Their multiple substitutions increase their affinity and lower their degradation rate compared to endogenous GnRHR, without activating the receptors. Their immediate action, while a benefit when time is limited, also comes with the disadvantage of requiring a constant presence in the blood stream, making long-term preparations necessary. Another disadvantage is their generally poor solubility and subsequent high dosing concentrations [30].

For both of the above, results are still inconclusive of the extent that they may aid fertility when administered before or during chemotherapeutic and/or radiotherapeutic treatment [34]. It does not seem true that the hormonal changes they create have a direct protective effect on the ovaries [35] and it rather appears to be primarily through the suppression of ovarian function.

## 4. Anti-Müllerian Hormone as an Estimator of Ovarian Reserve

AMH is produced by the primary, secondary, pre-antral and small antral follicles up to 8 mm in diameter. Larger antral follicles (more than 8 mm) in diameter do not produce AMH [36]. Thus, it is produced by all pre-antral follicles and early antral follicles, except for the primordial ones. As such, it is a marker of ovarian reserve and a predictor of quantitative response to controlled ovarian stimulation.

There are some reasons for using AMH as an ovarian reserve marker: it is not menstrual cycle dependent, with only small fluctuations occurring throughout it [37]. However, it may be influenced by the usage of oral contraceptives, which may lower AMH levels [38].

Limitations to AMH also exist. When it comes to cancer, AMH has only recently begun to be studied, with most studies focusing on breast cancer. Studies that look specifically at GnRHa co-treatment and its effect on AMH levels remain limited [39,40,41,42,43]. Pre-treatment AMH, combined with age, the other fundamental predictor, is instead an important marker to be evaluated during counselling. The efficacy of every fertility preservation method, including GnRHa, depends on the woman’s age, ovarian reserve, type and cumulative dose of the gonadotoxic therapy. Post-treatment AMH has limited utility as a predictor of menstrual restoration/fertility and currently cannot serve as a predictor of time to menopause [44].

## 5. Rationale of Using GnRH Analogues in Fertility Preservation Post Cancer Treatment

The use of GnRH analogues in order to achieve reduction of ovarian toxicity is based on the observation that chemotherapy mostly affects tissues with rapid cellular turnover, such as gonadal ones [26]. It also based on the fact that gonadotoxicity is lower in prepubertal girls than in adult women [14,27]. The latter could be because of their higher ovarian reserve, in addition to the hypogonadotropic prepubertal milieu. This could be because of a decrease in the proliferation rate of granulosa cells and a suppression of follicular recruitment, as GnRHas seem to stimulate the prepubertal hypogonadotropic milieu. Potential mechanisms for ovarian protection could be: (a) a reduction in ovarian blood flow via a direct effect on GnRH receptors that causes a decrease in the amount of chemotherapeutics that reach the ovary [45,46], (b) via a direct effect on ovaries such as up-regulation of intra-ovarian anti-apoptotic molecules and protection of germ line stem cells [28,47] and (c) indirectly by having an anti-apoptotic event on surrounding cumulus cells [48], as has been recently been stipulated.

Based on their mode of action, there are two reasons that we believe GnRH analogues could be used for fertility preservation. First, because of their fast-acting effects as established above. Secondly, because of their mechanism of action, as their suppressive ovarian effects may protect the oocytes from toxicity, making them beneficial in chemotherapeutic treatments such as alkylating agents and anthracyclines in adolescent girls and pre-menopausal women with ages between 15 and 45 [11].

## 6. GnRH Agonists and Fertility Preservation after Cancer Treatment

Up to date, over 50 publications (14 RCTs, 25 non-RCTs, and 20 meta-analyses) have reported on over 3100 patients during chemotherapy, receiving concurrently GnRH agonists for preservation of ovarian function via temporary ovarian suppression. These patients were treated for breast cancer, hematologic cancers, or autoimmune diseases. The above studies reported that the GnRHa adjuvant co-treated patients resumed regular menses and normal ovarian function in about 85% to 90% of cases as compared to the 40% to 50% in the chemotherapy only group. Furthermore, natural pregnancy rates in survivors who were co-treated with GnRHa adjuvant during gonadotoxic chemotherapy ranged from 23% to 88%, as compared to the 11% to 35% (*p* < 0.05) in control patients who were not co-treated [39,40,49,50,51,52,53,54,55,56,57,58,59,60,61,62]. More specifically, a long-term follow-up analysis (up to 15 years) of adolescent and young adults with Hodgkin’s lymphoma co-treated with triptorelin confirmed the gonadoprotective effect of GnRHa [63].

Indeed, 96.9% in the GnRHa group resumed ovulation and regular menses, throughout a median follow-up of 8 years (range 2–15), compared with 63% in the control group. Recently, a prospective non-randomized study in adolescent and young women treated for cancer compared the rate of POF after hematopoietic stem cell transplantation in those receiving GnRHa with gonadotoxic chemotherapy vs. chemotherapy alone [64]. The study found that GnRHa co-treatment significantly decreased the POF rate from 33% to 82%. Moreover, a recent single-center retrospective study on postmenarchal adolescent patients (median age 14, range 11 to 18) treated for acute lymphoblastic leukemia, acute myeloid leukemia, Hodgkin’s lymphoma, and other cancers showed that co-treatment with GnRH analogues preserved ovarian function and fertility in adolescents [65]. Other large retrospective and prospective studies, as well as case series, also showed a potential protective effect of GnRHa during chemotherapy in women with hematological malignancies [40,61,63,65,66,67,68].

Thus, the German Hodgkin Study Group HD14 trial analysis with 263 patients revealed that prophylactic use of GnRH analogues as a highly significant prognostic factor for preservation of fertility favoring pregnancies [40] in early Hodgkin’s Lymphoma patients after chemotherapy treatment. In addition, in another study where fertility status was assessed among 108 females of reproductive age treated by chemotherapy for newly diagnosed Hodgkin’s lymphoma between 2005 and 2010, authors concluded that chemotherapy with GnRH analogues used in more advanced Hodgkin’s Lymphomas retained ovarian function significantly better after two years [66].

On the contrary, randomized trials performed in women with hematological malignancies showed no GnRH analogue induced protective effect, nor suggested a partial protective effect, with only a delaying in the appearance of POF. All these studies had a small sample size and were not powered to find a possible advantage of GnRH analogues [41,42,68,69,70].

Thus, a study investigated the impact of leuprolide on ovarian function (Follicle stimulating hormone (FSH) levels) after myeloablative conditioning on 17 women undergoing hematopoietic cell transplant and concluded that leuprolide may protect ovarian function after myeloablative conditioning as 3 out of 7 evaluable Leupron recipients had ovarian failure 703 days post-transplant [68].

In a second study, they evaluated the best method to assess the ovarian reserve by measuring FSH, Luteinizing hormone (LH), inhibin B, AMH levels and the ultrasound antral follicular count in 29 women with Hodgkin’s Disease treated with chemotherapy. A combination of ultrasound antral follicular count and AMH levels were the best predictor of ovarian reserve. They concluded that GnRH analogue treatment did not have any protective effect but could delay the development of ovarian failure [41].

Similarly, another study reported the 5-year follow-up results on ovarian reserve, measured with AMH or FSH levels, of 67 patients with lymphoma randomly assigned to receive either triptorelin plus norethisterone or norethisterone alone during chemotherapy. They reported that AMH and FSH levels were similar in both groups while 53% and 43% achieved pregnancy in the GnRH analogues and control groups (*p* = 0.467) [70].

A clinical practice guideline by ASCO on ovarian suppression adjuvant endocrine therapy for women with HR+ breast cancer [71] stated that the addition of ovarian suppression to standard adjuvant therapy with tamoxifen or with an aromatase inhibitor improved DFS, disease, and distant recurrence, compared with tamoxifen alone. The panel concluded that high-risk patients should receive co-treatment with GnRHa to achieve ovarian suppression, in addition to adjuvant endocrine therapy. Thus, the results of all these publications implied that GnRHa might either improve or not affect the survival of patients receiving chemotherapy [28].

Regarding endometrial cancer, recently there was a small monocentric retrospective study in patients with early-stage endometrial cancer using a combination of surgery and GnRH agonist with a 3-month follow-up interval with endometrial sampling by hysteroscopy. It was concluded that GnRHas after surgery are an effective fertility-sparing strategy for women with grade 1 endometrial carcinoma and/or endometrial intra-epithelial neoplasia [72].

The only prospective phase III RCT including postmenarchal adolescent patients affected by ovarian malignancy demonstrated the gonadoprotective effect of GnRHa even in the younger population [73]. Six months after chemotherapy, all the patients in the GnRHa group had normal menstrual bleeding and normal titre of FSH/LH, whereas 33% in the control group had amenorrhea and POF.

On the other hand, there are in vitro studies that do not support the beneficial effect of GnRH analogues in fertility preservation post-chemotherapy. An in vitro study, using (*n* = 15 age = 14–37) human granulosa cells and ovarian tissue fragments expressing GnRH receptors, found that GnRH agonists administered with chemotherapy (e.g., cyclophosphamide, paclitaxel, fluorouracil, or a TAC (docetaxel, doxorubicin, cyclophosphamide) regimen) for 24 h neither activated anti-apoptotic pathways nor prevented follicle loss or DNA damage caused by the chemotherapeutic agents [43]. In the study, however, the administration of the GnRH agonists occurred concomitantly with the initiation of chemotherapy rather than approximately one week earlier (the minimal time required for ovarian suppression following the flare-up effect). Therefore, there is a chance that the initiation of chemotherapy concurred with the flare-up period of the GnRH agonist, potentially neutralizing the protective effect. The authors concluded that GnRH agonist treatment with chemotherapy does not prevent or ameliorate ovarian damage and follicle loss in vitro.

As also shown above, there are studies reporting on the effects of GnRH analogues on AMH levels. One study included 263 women with early-stage HL who all received GnRH analogues treated either with less gonadotoxic chemotherapeutic agents (Adriamycin, Bleomycin, Vinblastine, Dacarbazine), also known as the ABVD regimen, or with more aggressive alkylating agents, such as the BEACOPP regimen (bleomycin, etoposide, adriamycin, cyclophosphamide, vincristine, procarbazine, prednisone), found that FSH and AMH hormonal levels were significantly better in the ABVD plus GnRH analogues arm, one year post treatment [40]. In another human study, studying 84 patients diagnosed with Hodgkin’s or non-Hodgkin’s lymphoma who completed the one-year follow-up after being treated with chemotherapy and GnRH analogues, it was reported that the group receiving GnRHa co-treatment had a significantly higher proportion of AMH values with >1 ng/mL compared to the control group (8/16 vs. 2/15; *p* = 0.023), as well as significantly higher mean AMH values (1.40 ± 0.35 vs. 0.56 ± 0.15 ng/mL; *p* = 0.040) [39]. However, the small sample size of 16 and 15 patients of the GnRH and control groups respectively limits the significance of this positive result. Another study, however, evaluating patients treated for Hodgkin’s disease, found no discernible difference between AMH levels of the GnRH co-treated group and control group [41].

Its findings agree with a study that investigated the use of oral contraceptives and GnRH agonists as co-treatment during advanced HL chemotherapeutic treatment, where AMH levels remained practically below detection levels for all patients [42]. An in vitro study found in the control group without chemotherapy or GnRH analogue, AMH was indeed correlated with the number of growing follicles. As soon as chemotherapy was introduced, however, any correlation disappeared [43]. It should be noted that due to the general toxicity to any growing follicles during early stages after chemotherapy, we would not expect to see noticeable AMH levels for at least a few months post-treatment (Table 2).

## 7. GnRH Antagonists and Fertility Preservation Post Cancer Treatment

There are limited data regarding the effectiveness of GnRH antagonists for fertility preservation in gynecological cancer. Most are small animal studies and there is a general lack of human data.

An animal study assessed whether a GnRH antagonist ((GnRHant); in this study cetrorelix) was able to protect ovaries from chemotherapy damage in 42 female Wistar rats. The rats were divided into four groups: group I (*n* = 9) received placebo; group II (*n* = 12) received placebo+cyclophosphamide (CPA); group III (*n* = 12) received GnRHant+CPA; and group IV (*n* = 9) received GnRHant+placebo. The estrous cycle was studied using smears, pregnancies were documented, the number of live pups measured, and the ovarian cross-sectional area was measured, together with follicle count. The ovarian cross-sectional area was not different between groups, neither was the number of individual follicle types. However, rats on GnRH antagonists and placebo (Group IV) had a higher total number of ovarian follicles than those in the control group. Researchers conclude that the use of a GnRH antagonist before CPA chemotherapy provided fertility protection [75] (Table 3).

## 8. Combination of GnRH Agonists and Antagonists and Fertility Preservation Post Cancer Treatment

To date it is already known that both GnRH agonists and antagonists have disadvantages that limit their use; GnRHas causes a flare-up effect during the first week after administration and no long-acting GnRHant agent is available. GnRHas combined with GnRHants may prevent the flare-up effect of GnRHa and rapidly inhibit the female gonadal axis. A small number of experimental animal studies with small sample sizes have reported controversial conclusions.

In a study involving 30 female Sprague Dawley rats of adolescent age, rats were randomized into five treatment groups (*n* = 6/group): (1) placebo, (2) cyclophosphamide (CPA) alone, (3) GnRH antagonist followed by GnRH agonist with placebo, (4) GnRH antagonist followed by GnRH agonist with CPA, and (5) GnRH agonist with CPA. The main outcome measure was live birth rate (LBR), and secondary measures included rat weight, ovarian volume, and follicles. Group 2 had decreased LBR. Group 4 and 5 had LBR similar to placebo. Ovarian volume did not vary between the groups. The CPA-alone group had fewer antral follicles compared to the control. The study demonstrated that the combination of GnRH antagonist and GnRH agonist and GnRH agonist alone preserved fertility in female adolescent rats following gonadotoxic chemotherapy treatment [76].

In another controlled animal study, researchers investigated the advantages of combination treatment with GnRHas and GnRHants in rats aged 12 weeks. The combination of a GnRH agonist with an antagonist completely prevented the flare-up effect and protected primordial ovarian follicles in the rats’ ovary from cisplatin-induced gonadotoxicity [77].

Furthermore, in a control experimental animal study, the aim was to assess the ovarian reserve with AMH and perform histology analysis after exposure to cisplatin with a GnRHa or GnRHant. Twenty-four Wistar albino rats were randomly divided into three groups. In group 1, rats received a single dose of 50 mg/m^2^ cisplatin with 1 mg/kg triptorelin. In group 2, rats received a single dose of 50 mg/m^2^ cisplatin with 1 mg/kg cetrorelix. In the control group (group 3), rats received 50 mg/m^2^ cisplatin. AMH levels and histology were used to assess ovarian reserve. Primary follicle counts were higher in group 2 whereas secondary follicle counts were higher in group 1. Both groups 1 and 2 had higher numbers of tertiary follicles and AMH levels than the control group [78] (Table 4).

## 9. Discussion

In this review, we explored the available data on the use of GnRH analogues as a co-treatment with chemotherapy in order to reduce gonadotoxicity in premenopausal patients with cancer. It has been hypothesized that ovarian suppression may have some gonadoprotective effects during gonadotoxic therapy. A potential mechanism for ovarian protection could be a reduction in ovarian blood flow that causes a decrease in the amount of chemotherapeutics that reach the ovary. Indeed uterine blood flow has been shown to be reduced after administration of GnRH analogues, although other studies did not detect difference [79]. Another two potential mechanisms are a decreased rate for granulosa cell proliferation and a suppression of follicular recruitment. These last two are based on the observation that chemotherapy mostly affects tissues with rapid cellular turnover, like gonads [26], and thus the gonadotoxicity is lower in prepubertal girls than adult women [27]. An alternative explanation is because of their higher ovarian reserve in addition to the hypogonadotropic prepubertal milieu. Thus, GnRH agonists seem to stimulate the prepubertal hypogonadotropic milieu, to have direct effect on GnRH receptors, to decrease ovarian perfusion [47], and act directly on ovaries through up-regulation of intra-ovarian anti-apoptotic molecules and protection of germ line stem cells [28,47].

Most studies (mentioned in Section 7, Section 8 and Section 9 and Table 2, Table 3 and Table 4) support the findings that GnRH agonist co-treatment protects from gonadotoxicity and preserves fertility in chemotherapy-treated pre-menopausal women with breast cancer and hematological malignancy. Specifically, most studies supporting GnRH agonists as a co-treatment in the cases of premenopausal cancer chemotherapy for fertility preservation, refer to breast cancer or to hematological malignancy. There are no large studies available regarding the possible fertility preservation effect of GnRH agonist co-treatment in the cases of premenopausal women treated with chemotherapy due to ovarian, endometrial, or cervical cancer, and thus we have inconclusive data. An explanation to the above could be that most of these patients present at a later age and have completed their family. In addition, for young women with cervical cancer the most accepted method for fertility preservation is fertility preserving surgery (i.e., radical trachelectomy), in highly selected cases with transposition of the ovary. Gonadotoxic chemotherapy is rarely used for endometrial cancer. For ovarian cancer, fertility-sparing surgery has been applied in a very selected group of patients with Stage IA disease grade 1 that did not require chemotherapy. Existing guidelines (Table 1) state that GnRH agonists can be offered to women with breast cancer and potentially other cancers for the purpose of protection from ovarian insufficiency. They do not refer to the use of GnRH analogues for fertility preservation in women with hematological malignancies post-chemotherapy. Furthermore, they state that GnRH analogues should not replace oocyte/embryo cryopreservation as the established modalities for fertility preservation.

Regarding the use of GnRH antagonists, as a co-treatment with chemotherapy in gynecological cancer and hematological malignancies, data are not conclusive as there are only a few, limited, animal data. Our perspective on the above is that although there are possible mechanisms explaining the potential effects, there are some points that we further need to consider when examining possible benefits. Any potential beneficial effects of GnRH analogues as a co-treatment in fertility preservation could depend on the type and maybe the stage of cancer treated and possibly the type of alkylating agents used. The latter is based on the observation of the significant differences seen in fertility preservation of breast and hematological cancer compared to other gynecological breast cancers. In addition to that. age and/or ovarian reserve could be an important factor as females in pre-pubertal stage seem to be more protected. Furthermore, as the needed power to detect differences between the study results requires hundreds of patients in order to be able to come to safe conclusions, we need data from several large human studies. Lastly, studies need to be homogeneous regarding the fertility preservation criteria they use as outcomes. Researchers might need to clarify the criteria they use to study the effectiveness of GnRH analogue co-treatment in fertility preservation treatments for premenopausal patients with gynecological cancers. For example, not all studies consider ovarian reserve as a criterion of fertility preservation assessment and use instead pregnancy rates, live birth rates, and look at long-term fertility.

Basic future research could focus on investigating the differential effects of GnRH analogue co-treatment on the physiology of different ovarian cell populations. In particular, the potential antiapoptotic effect of GnRHas on the several types of follicular cells as well as in the mesenchymal stroma cells should be further investigated. Whereas GnRHRs have been identified in several cell lines in the ovary [80] their absence from pre-antral follicles per se [48] creates several questions as to the protective effect of GNRH analogues. Furthermore, the impact of decreased ovarian perfusion and thus decreased delivery of the cytotoxic agents to the ovary as protective mechanism should also be evaluated. Clinical research could focus on the effects of GnRHa co-treatment with chemotherapy: first by evaluating surrogate markers of ovarian reserve such as AMH before during and after gonadotoxic therapy, secondly by evaluating other markers of ovarian reserve that could be more accurate, and thirdly by presenting the actual impact of their use in women that attempt pregnancy after treatment.

Limited and conflicting results were found for AMH levels as a fertility preservation indicator after treatment with GnRH analogues. Apart from one study [39], others did not discern any impact on the use of GnRH analogues in AMH levels [40,41,42,43]. Invariably, AMH levels seem to fall to almost zero during chemotherapy regardless of treatment and the post-chemotherapy levels in the ASTRRA trial (82 participants) seem to be an accurate predictor (86.7%) of the recovery of ovarian function during resumption of menstruation in breast cancer patients [79]. Nonetheless for post-chemotherapy recovery of AMH levels, available data are inconclusive. A recent small study (50 patients) in premenopausal patients (<40 years old) with early breast cancer who received chemotherapy and co-treatment with GnRHa triptorelin reported that AMH decreased to nearly undetectable levels after chemotherapy and recovered after 12 months. It did not, however, exceed one tenth of the pre-treatment levels although 48% of patients recovered above a threshold of 0.2 ng/ml compared to those who did not have co-treatment [81].

In conclusion, studies so far support the use of GnRH agonists as a co-treatment in order to provide gonadal protection and subsequently fertility preservation in women with breast cancer and hematological malignancy in general. There is a paucity of data regarding other types of gynecological cancer. Nevertheless, data extrapolated from studies involving young patients with breast cancer supports a potential beneficial effect of the use of GnRH analogues during chemotherapy with no adverse oncological impact [40,82,83]. On the contrary, there are studies to support a small beneficial effect on survival and decrease-free interval with the co-administration of GnRH analogues during gonadotoxic therapy. Large human studies need to take into consideration age, stage, type, and treatment of cancer used, as well as fertility preservation assessment criteria. It seems that we might have to individualize GnRH co-treatments in patients treated for gynecological cancer. Indeed, as mentioned in the previous sections, numerous randomized trials, systematic reviews, and meta-analyses have shown a correlation of GnRH analogue use before and during chemotherapy with lower rates of premature ovarian insufficiency [74,83]. According to clinical practice guidelines, in most cases GnRH analogues do not protect ovaries from radiotherapy-induced gonadotoxicity and so they are not suggested for female patients scheduled to receive pelvic, abdominal, or total body irradiation [15,74,84].

## Figures and Tables

**Table 1 ijms-23-02287-t001:** International guidelines on GnRH analogues.

Guideline	Year of Publication	Recommendation	Methodology
ASCO [15]	2018	“There is conflicting evidence to recommend gonadotropin-releasing hormone agonists (GnRHa) and other means of ovarian suppression for fertility preservation. The Panel recognizes that when proven fertility preservation methods such as oocyte, embryo, or ovarian tissue cryopreservation are not feasible, and in the setting of young women with breast cancer, GnRHa may be offered to patients in the hope of reducing the likelihood of chemotherapy-induced ovarian insufficiency. However, GnRHa should not be used in place of proven fertility preservation methods.”	Systematic review of the literature, published from January 2013 to March 2017, was completed using PubMed and the Cochrane Library.
ASRM [23]	2019	“GnRH agonists can be offered to women with breast cancer and potentially other cancers for the purpose of protection from ovarian insufficiency. However, GnRH analogues should not replace oocyte/embryo cryopreservation as the established modalities for fertility preservation.”	Systematic reviews, meta-analyses and RCTs between the years 2006–2018.
ESHRE [22]	2019	“For ovarian stimulation in women seeking fertility preservation for medical reasons the GnRH antagonist protocol is probably recommended. There is moderate quality evidence of the necessity of considering a specific GnRH analogue protocol. GnRH antagonist protocols are preferred since they shorten the duration of ovarian stimulation, offer the possibility of triggering final oocyte maturation with GnRH agonist in case of high ovarian response, and reduce the risk of Ovarian Hyperstimulation Syndrome (OHSS). Moreover, especially in cancer patients, who are at higher risk of thrombosis due to their oncologic status, seem to be preferred since they enable GnRH agonist trigger, therefore reducing the risk of OHSS.”	The search was based on a final list of 18 key questions. Key words were sorted by importance and used for searches in PUBMED/MEDLINE and the Cochrane library. The search was performed up to 8 November 2018. Literature searches were performed as an iterative process. In a first step, systematic reviews and meta- analyses were collected.

**Table 2 ijms-23-02287-t002:** Studies on GnRH agonists & fertility preservation post cancer treatment.

Author	Study Design	Fertility Preservation	Discussion
Cuzick et al. [49]	Meta-analysis; 16 RCTs of 11,906 premenopausal women with early breast cancer. Data collected from 1987 to 2001. Published in 2007.	Ovarian suppression achieved for the majority of a goserelin study group (70%).	GnRH agonists slightly decreased the changes of pre-menopausal women developing permanent amenorrhea.
Lambertini et al. [50] Contains RCT [74]	Meta-analysis; 12 RCTs with a total of 1231 breast cancer patients. Data collected from 2008 to 2015. Published in 2015.	Significant reduction in POF cases for patients using GnRHas during chemotherapy (*p* < 0.001). A significant (*p* = 0.041) higher percentage of patients taking GnRH became pregnant post-treatment as compared to controls (9.2% vs. 5.5%).	Usage of GnRHas treatment reduces risk of chemotherapy induced POF in young women.
Del Mastro et al. [53] Contains RCTs [39,42,73]	Meta-analysis; nine studies of 765 pre-menopausal cancer patients. Data collected from 2007 to 2013. Published in 2014.	Significant reduction in the risk of POF in patients taking GnRHa before and during chemotherapy (OR = 0.43; *p* = 0.0130).	GnRHa ovarian suppression reduces chemotherapy induced POF risk in pre-menopausal patients. Protective effect similar across age groups and timing of POF assessment.
Chen et al. [54] Contains RCTs [41,55]	Meta-analysis; 12 RCTs of 1369 women ages 12–51.1 years old. Data collected from 1996–2012. Published in 2019.	Menstruation recovery/maintenance significantly higher in the GnRHa-taking group than the non-taking (74.5%vs. 50.0%) (*p* = 0.006) POF significantly lower in the co-treatment group than chemotherapy only (25.3% vs. 10.7%; *p* < 0.00001).	GnRH seems effective in continuation of menstruation, ovulation and reducing treatment related POF. Evidence for protection of fertility unclear.
Munhoz et al. [56]	Meta-analysis; 7 RCTs with 1047 pre-menopausal patients with early breast cancer. Data collected from 2009 to 2015. Published in 2016.	Higher rate of menses recovery at 6- and 12-months post-treatment with GrHa (*p* = 0.002, *p* < 0.001 respectively).	GnRHa co-treatment associated with increased rate of regular menses.
Behringer et al. [40]	1579 women patients of 8–60 year of age with HL stages 1–2B treated with ABVD. Data collected up to 2011. Published in 2012.	Prophylactic GnRHa usage highly effective for preservation of fertility (OR = 12.87; *p* = 0.001).	ABVD treatment with GnRHa treatment seems to preserve fertility.
Wong et al. [57]	125 pre-menopausal women with early breast cancer. Data collected up to 2009. Published in 2013.	84% of women recovered normal menstruation. 71% of women who attempted pregnancy conceived.	Ovarian toxicity usually seen with chemotherapy not observed when co-treated with GnRHa goserelin.
Recchia et al. [58]	42 pre-menopausal women with breast cancer and more than 10 positive auxiliary nodes. Data collected up to 2015. Published in 2015.	13 women resumed regular menses, three of which had four full-term pregnancies in total, post-chemotherapy.	Moderate toxicity and ovarian function preserved; improved expected DFS and OS rates.
Recchia et al. [59]	200 pre-menopausal women patients with high-risk early breast cancer. Data collected up to 2007. Published in 2015.	After median 105-month follow-up, no woman <40 years old exhibited POF, 44% of women >40 years old did. DFS and OS rates 85% and 91% respectively.	GnRHa co-treatment in adjuvant chemotherapy prevented POF and was linked with improved DFS and OS rates.
Blumenfeld et al. [60]	95 women undergoing chemotherapy before stem cell transplantation. Data collected up to 2008. Published in 2012.	GnRHa co-treatment had a significant effect in increasing cyclical ovarian function rates than without (66.7% vs. 18.2%; *p* = 0.02) for patients with lymphomas. No significant change for leukemia patients taking GnRHa than without (10.0% vs. 8.3%; *p* = 1).	Gonadotoxicity and POF may be significantly decreased with GnRHa for lymphoma patients. Results for leukemia patients inconclusive.
Blumenfeld et al. [61]	Follow-up on a woman that delivered two neonates, years after stem cell transplantation therapy, which on its own inevitably leads to POF. Patient had co-treatment with GnRHa. Data collected up to 2008. Published in 2010.	Patient spontaneously delivered 11- and 12-years post SCT treatment with chemotherapy with GnRHa co-treatment.	GnRHa may have minimized gonadotoxic effect to the point of maintaining fertility.
Moore et al. [62]	218 pre-menopausal women with stage I-IIIA estrogen receptor negative, progesterone receptor negative breast cancer. Data collected up to 2011. Published in 2019.	Significant increase in pregnancies post therapy in patients that received GnRHa than without (OR = 2.34; *p* = 0.03). Disease free survival and overall survival rates were not significantly different.	Patients undergoing GnRHa more likely to avoid POF and more likely to conceive.
Blumenfeld et al. [63]	65 women patients receiving monthly GnRHa injections during chemotherapy compared to 46 women control group who were not. Data collected up to 2005. Published in 2008.	96.9% treated with GnRHa resumed menses versus 63% in control.	GnRHa co-treatment significantly reduces ovarian failure for patients treated for HL.
Meli et al. [64]	Retrospective observational study: 36 pre-menopausal cancer patients co-administered with GnRHa. 9 of these patients underwent hematopoietic stem cell transplantation (HSCT). Data collected up to 2015. Published in 2018.	Non-HSCT cases (27) all maintained normal ovarian function.	GnRHa co-treatment prevented POF in nonHSCT cases; it was not effective at preserving ovarian function in the cases with HSCT.
Gini et al. [65]	97 pre-menopausal women with Hodgkin’s and non-Hodgkin’s lymphoma undertaking chemotherapy with or without GnRHa co-treatment. Data collected up to 2012. Published in 2019.	Resumption of regular menses associated with the usage of GnRHas (*p* = 0.034).	GnRHas may have a protective effect against gonadotoxicity in chemotherapy for Hodgkin’s and non-Hodgkin’s lymphoma.
Huser et al. [66]	108 pre-menopausal patients treated for HL, all co-treated with GnRHas. Data collected up to 2010. Published in 2015.	Two years post-treatment 90.7% of patients retained ovarian function and 21.3% achieved clinical pregnancy.	Higher ovarian function retainment associated with GnRHa co-treatment.
Blumenfeld et al. [67]	Retrospective cohort study: comparison of 261 patients with GnRHa co-treatment vs. 188 patients who were treated with chemotherapy alone. Data collected up to 2015. Published in 2015.	Significant higher clinical ovarian function rates in co-treatment patients than without (87% vs. 49% OR = 6.8; *p* = 0.0001). Higher chance of spontaneous pregnancies compared to control group (65.6% vs. 37.97: 0.0004).	GnRHa co-treatment significantly increases COF.
Phelan et al. [68]	19 women observed, 9 of which underwent hematopoietic cell transplantation (HCT) co-treated with GnRHa, the others without. Data collected up to 2014. Published in 2016.	57% of the co-treated group experienced POF, a much lower rate than the historic average of 90%.	GnRHa leuprolide appears to preserve ovarian function in HCT patients.
Waxman et al. [69]	17 women were split in a control and study group given GnRH prior to and during chemotherapy. Data collected up to 1987. Published in 1987.	50% of the study group became amenorrhoeic (4/8), vs. 66% in the control group control (6/9).	GnRHa buserelin was not significantly effective at preserving fertility.
Demeestere et al. [70]	129 lymphoma patients randomly assigned to receive GnRHa co-treatment or not. Data collected up to 2010. Published in 2016.	In a five-year follow-up, co-administration with GnRHa did not seem to be correlated with reduced POF risk. Pregnancy rates were similar in the two groups (53% rate in GnRHa, 43% in control; *p* = 0.467).	GnRHa co-treatment was not found to be an effective fertility preservation tool in young patients with lymphoma.
Tock et al. [72]	Retrospective review: 18 pre-menopausal women with grade 1 endometrial carcinoma (G1EC) and/or endometrial intraepithelial neoplasia (EIN), all of which received GnRHa combined endometrial resection and laparoscopy. Data collected up to 2016. Published in 2018.	12 patients conserved their uterus, eight patients became pregnant with 14 pregnancies among those who tried to become pregnant.	GnRHa is an effective fertility preserving option compared to other treatments for G1EC and EIN.
Bildik et al. [43]	15 ovarian cortical pieces, mitotic non-luteinized and non-mitotic luteinized granulosa cells expressing GnRH receptor were treated with chemotherapeutic agents, with or without GnRHa. Data collected up to 2015. Published in 2015.	GnRHa samples compared to control raized intracellular cAMP levels but did not activate any anti-apoptotic pathways nor prevented follicle loss.	GnRHa co-treatment does not prevent or alleviate ovarian damage and follicle loss in vitro.

**Table 3 ijms-23-02287-t003:** GnRH antagonists only & fertility preservation during cancer treatment.

Author	Study Design	Results	Discussion
Lemos et al. [75]	42 female Wistar rats treated in four different groups: placebo or cyclophosphamide, GnRHa antagonist or placebo. Data collected up to 2010. Published in 2010.	Rats in the group that received GnRHant treatment had a higher number of total follicles than the control group (*p* < 0.05).	GnRHant treatment before chemotherapy resulted in some fertility protection in rats.

**Table 4 ijms-23-02287-t004:** Combination of GnRH agonists and antagonists and fertility preservation post cancer treatment.

Author	Study Design	Results	Discussion
Knudtson et al. [76]	30 female rats in groups of six each with either placebo, cyclophosphamide, GnRHa + GnRHant + placebo, GnRHa + GnRHant + cyclophosphamide or GnRHa + cyclophosphamide. Data collected up to 2016. Published in 2017.	The combined approach + cyclophosphamide vs. GnRHa + cyclophosphamide did not have any significant differences on average birth rates (12.8 ± 2.7 vs. 12.3 ± 1.6).	The addition of a GnRHant to a GnRHa did not seem to provide a greater protective effect.
Li et al. [77]	72 Rats aged 12 weeks received chemotherapy either with GnRHa, GnRHant, or combination. Data collected up to 2013. Published in 2013.	Long-term combination provided the largest percentage of normal menstrual cyclicity return vs. antagonist alone or agonist alone (66.7%, 33.3%, 25.0% respectively).	Combination treatment prevented flare-up effect.
Tas et al. [78]	24 Winstar albino rats divided into three groups, receiving chemotherapy either with GnRHa (group 1), GnRHant (group 2), or without (group 3). Data collected up to 2019. Published in 2019.	Total follicle count was higher in group 1 and 2 than control (14.32 ± 5.96 vs. 12.48 ± 4.12 vs. 10.63 ± 6.80).	GnRHa and GnRHant displayed protective effects against cisplatin gonadotoxicity in rats.

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
