# Peer review of "GnRH Analogues as a Co-Treatment to Therapy in Women of Reproductive Age with Cancer and Fertility Preservation"

_ijms, 2022, doi:10.3390/ijms23042287_

Round 1
Reviewer 1 Report
Thanks for adding my suggestion.
I think this manuscript is improved.
Author Response
We thank the reviewer for his comment.
Reviewer 2 Report
Dear authors,
the review article "GnRH analogues as a co-treatment to therapy in women of reproductive age with cancer and fertility preservation" deals with an important issue. The article is informative and well-structured. There are some minor language corrections which should be performed. For example, in the abstract it should be "as a prophylaxis" instead of prophylaktisch in line 3. There are also some other minor language issues which should be revised.
Beides, you should additionally cite the article " Fertility preservation in female cancer patients: current knowledge and future perspectives" published in Minerva Ginecologica in 2019 by Findeklee et al. in your references. It provides basic information on both, fertility preservation in general as well as GnRH agonists.
Afterwards, your article should be ready to get published.
